# A Bacterial Platform for Studying Ubiquitination Cascades Anchored by SCF-Type E3 Ubiquitin Ligases

**DOI:** 10.3390/biom14101209

**Published:** 2024-09-25

**Authors:** Zuo-Xian Pu, Jun-Li Wang, Yu-Yang Li, Luo-Yu Liang, Yi-Ting Tan, Ze-Hui Wang, Bao-Lin Li, Guang-Qin Guo, Li Wang, Lei Wu

**Affiliations:** 1Ministry of Education Key Laboratory of Cell Activities and Stress Adaptations, School of Life Sciences, Lanzhou University, Lanzhou 730000, China; puzx21@lzu.edu.cn (Z.-X.P.); jlw@lzu.edu.cn (J.-L.W.); lyuyang2023@lzu.edu.cn (Y.-Y.L.); liangly2018@lzu.edu.cn (L.-Y.L.); tanyt2023@lzu.edu.cn (Y.-T.T.); wzehui2023@lzu.edu.cn (Z.-H.W.); libl20@lzu.edu.cn (B.-L.L.); gqguo@lzu.edu.cn (G.-Q.G.); 2Gansu Province Key Laboratory of Gene Editing for Breeding, School of Life Sciences, Lanzhou University, Lanzhou 730000, China; 3School of Chemical Engineering Ocean and Life Science, Dalian University of Technology, Panjin 124221, China

**Keywords:** *Duet* expression system, co-expression recombinant proteins, detection of ubiquitination, auto-ubiquitination of TIR1, AUX/IAAs

## Abstract

Ubiquitination is one of the most important post-translational modifications in eukaryotes. The ubiquitination cascade includes ubiquitin-activating enzymes (E1), ubiquitin-conjugating enzymes (E2), and ubiquitin ligases (E3). The E3 ligases, responsible for substrate recognition, are the most abundant and varied proteins in the cascade and the most studied. SKP1-CUL1-F-Box (SCF)-type E3 ubiquitin ligases are multi-subunit RING (Really Interesting New Gene) E3 ubiquitin ligases, composed of CUL1 (Cullin 1), RBX1 (RING BOX 1), SKP1 (S-phase Kinase-associated Protein 1), and F-box proteins. In vitro ubiquitination assays, used for studying the specific recognition of substrate proteins by E3 ubiquitin ligases, require the purification of all components involved in the cascade, and for assays with SCF-type E3 ligases, additional proteins (several SCF complex subunits). Here, the *Duet* expression system was used to co-express E1, E2, ubiquitin, ubiquitylation target (substrate), and the four subunits of a SCF-type E3 ligase in *E. coli*. When these proteins co-exist in bacterial cells, ubiquitination occurs and can be detected by Western Blot. The effectiveness of this bacterial system for detecting ubiquitination cascade activity was demonstrated by replicating both AtSCF^TIR1^-mediated and human SCF^FBXO28^-mediated ubiquitylation in bacteria. This system provides a basic but adaptable platform for the study of SCF-type E3 ubiquitin ligases.

## 1. Introduction

Ubiquitin (Ub) is a highly conserved protein composed of 76 amino acids. A single Ub (monoubiquitination) or Ub chains (polyubiquitination) can be conjugated to proteins. Ubiquitination is a major regulatory mechanism for protein degradation and impacts functions such as DNA repair, transcription, cell differentiation, cell cycle regulation, and stress responses [1]. The covalent attachment of ubiquitin or ubiquitin chains to the target protein requires a multi-step, ATP-dependent process controlled by ubiquitin-activating enzymes (E1s), ubiquitin-binding enzymes (E2s), and ubiquitin ligases (E3s), which is also known as the ubiquitination cascade. The cascade begins with an E1 forming a thioester bond with the C-terminal carboxyl group of ubiquitin. Next, the thioester-linked ubiquitin is transferred to a cysteinyl residue of an E2 enzyme. Finally, the E3 ubiquitin ligase binds directly to the substrate and catalyzes the transfer of Ub from E2 to a lysine residue of the target protein [2,3].

E1’s initiation of the coupling cascade has little effect on target specificity. There are only two E1 subtypes in *Arabidopsis thaliana*, one of which may be nuclear-localized [4]. In contrast, the E2 enzymes comprise a large family, including at least 37 E2 genes, which are divided into 12 subfamilies and eight E2-like genes in the *A. thaliana* genome [5,6,7]. In the ubiquitination cascade, E3 is responsible for recruiting target proteins for ubiquitination and is the major substrate recognition component of this pathway. The number of E3 is particularly large in eukaryotes, with 1400 predicted in *A. thaliana* [2]. Based on the presence of conserved domains and the mechanism of ubiquitin transfer to the substrate protein, E3s can be divided into four groups: RING (Really Interesting New Gene), U-box, HECT (Homologous to the E6AP C-Terminus), and RBR (RING-between RING) [8].

Among the four groups of E3 ligases, RING E3s are the most abundant in eukaryotes. The characteristic of this group is the presence of a Ring-Finger domain. The Ring-Finger domain contains 8eightconserved amino acid residues (Cys or His) that coordinate two zinc ions, which binds to an E2-ubiquitin conjugate to transfer ubiquitin to a lysine residue of the target protein [9,10]. Based on their subunit composition and mechanism of action, the RING E3 group can be further divided into several subgroups, including monomer RING E3s (that can function as monomers, homodimers, or heterodimers), Cullin-RING Ligases (CRLs), and the anaphase-promoting complex [8,11,12].

CRLs are the most widespread class of E3s in eukaryotes. Studies to date have shown that all eukaryotes contain a family of Cullin proteins and that homologs for most Cullin proteins can be identified across kingdoms. CRLs consist of several subunits. The Cullin subunit serves as a scaffold, recruiting a RING-Finger domain containing the RBX1 (RING BOX 1) subunit to its C-terminal region and binding one or more additional Cullin-specific subunit(s), which serve as substrate receptors and as adaptors, to its N-terminus [12]. There are three types of CRLs in plants, each built on a different Cullin: CUL1, CUL3, and CUL4. The CUL1-scaffolded E3 ligases are the most intensively studied CRLs due to their important function in plants [13].

For CUL1, the adaptor subunit is SKP (S-phase Kinase-associated Protein) [10,14], which together form SCF (SKP1-CUL1-F-Box) complexes. Within the SCF complex, the SKP adaptor protein binds an F-box protein that is responsible for substrate recognition and determines the specificity of the SCF complex, so the name of the F-box subunit is written in superscript to distinguish between different SCF-type E3 ligases. SCF-type E3 ligases are a highly flexible class of ubiquitin ligases. Within the proteomes of different species, a large number of F-box and SKP proteins is present, suggesting a conserved combinatorial organization of the SCF complex. The same CUL1-RBX1 works with different F-boxes and SKPs to mediate the ubiquitination of different target proteins [13].

While studying ubiquitination, researchers face several major challenges: (1) The high redundancy of components in the ubiquitination cascade (i.e., redundancy between E2/E3s and their ubiquitylation targets) makes it difficult to determine the connections between them in vivo; (2) The instability of ubiquitinated proteins in cells due to rapid deubiquitylation or degradation by the 26S proteasome, which makes them difficult to detect; and (3) The required activation of ubiquitination modifications in eukaryotic cells, which usually occur as a component of cellular signaling under specific circumstances [15]. For these reasons, in vitro experiments are often used to reconstruct ubiquitination cascades for research. However, many E3 ligases and their substrates are unstable and difficult to purify in their full-length forms. Moreover, ubiquitination is a highly dynamic and complex biochemical process involving the interaction of multiple enzymes and proteins. Reconstructing ubiquitination in the tube requires not only maintaining the activity of these components but also their interactions, which is very challenging under experimental manipulation [15]. For the multi-subunit E3 ubiquitin ligase SCF complex, it is even more difficult to reproduce the ubiquitination cascade in vitro. To make in vitro experiments easier to perform, some researchers developed a bacterial ubiquitination system for detecting the ubiquitination by singular E3 ligases to their particular substrates [15,16,17,18,19]. Co-expressing components of the entire ubiquitination cascade in *Escherichia coli* has successfully reproduced the ubiquitination cascade in a bacterial system. In addition to its convenience and ease, the other advantage of this method is that the ubiquitination tagging for degradation does not naturally exist in *E. coli* cells, and therefore stable ubiquitinated proteins can accumulate, which facilitates subsequent studies [15,16]. However, to date, there are no reports of this system being used for multi-subunit E3 ubiquitin ligases. Here, we extend the use of this bacterial system to ubiquitin cascades containing the multi-subunit E3 ubiquitin ligase SCF complex. When CUL1, RBX1, SKP1, and F-box protein are co-expressed with other components of the ubiquitination cascade, ubiquitination can occur in *E. coli* cells. In this paper, we selected AtSCF^TIR1^ and the human HsSCF^FBXO28^ as representatives of SCF-type E3 ligases to evaluate this system.

## 2. Materials and Methods

### 2.1. Plasmid Design for the Expression of Proteins Associated with the Ubiquitination Cascade

The *Duet* expression system was used to co-express the E1, E2, ubiquitin, ubiquitylation target (substrate), and four subunits of an SCF-type E3 ligase in *E. coli*. The original plasmids *pETDuet-1* (Cat. NO. 71146), *pACYCDuet-1* (Cat. NO. 71147), *pCDFDuet-1* (Cat. NO. 71340), and *pRSFDuet-1* (Cat. NO. 71341) were ordered from MERCK (Darmstadt, Germany). Details of the plasmids can be found on “https://www.merckmillipore.com/ (assessed on 25 September 2024)”.

The genes encoding the ubiquitination proteins of *A. thaliana*, namely *AtUBA1* (Ubiquitin Activating Enzyme 1), *AtUBC8* (Ubiquitin Conjugating Enzyme 8), *AtUBQ11* (Ubiquitin 11), *AtRBX1*, *AtASK1* (Arabidopsis SKP1 Homologue 1), and *AtCUL1*, were cloned into a variety of compatible Duet expression vectors, each with a different antibiotic resistance selective marker. In order to better detect these proteins in this experimental system, we changed the coding sequences corresponding to the original short tagging peptides in some of the Duet expression vectors. Three plasmids, *pCDFDuet-AtUBC8-S-AtUBA1-S*, *pRSFDuet-His-Flag-AtUBQ11*, and *pETDuet-AtRBX1-T7-AtASK1-T7-AtCUL1-S*, were constructed as shown in (Figure 1A).

The coding sequences of *AtUBC8* (E2) and *AtUBA1* (E1) in the first and second Multiple Cloning Sites (MCSs) of the *pCDFDuet-1* vector, respectively, had an S-Tag fused at the C-end of each of them. *AtUBQ11* encodes a Ub polymer with three repeats of the 228-bp coding sequence connected head to tail. One 228-bp fragment was cloned at the C-terminal of a His-Flag tag in the *pRSFDuet-1* vector. In vector *pETDuet-1*, there are only two MCSs for the expression of exogenous proteins, but we wanted to simultaneously express three subunits of the SCF complex (AtRBX1, AtASK1, and AtCUL1) using a single vector. To accomplish this, an open reading frame containing two cistrons was designed. The coding sequences of *AtRBX1-T7* and *AtASK1-T7* with stop codons were cloned in tandem with an RBS (ribosome binding site) added in front of the *AtASK1-T7* sequence. This cassette was inserted into the first MCS of the *pETDuet-1* vector. Next, the coding sequence of *AtCUL1* was inserted into the second MCS of the *pETDuet-1* vector.

Each of the three constructed vectors was independently transfected into *E. coli* strain BL21(DE3). After cultivation and induction with IPTG, the crude protein was extracted. Protein-specific antibodies were used for the Western Blot (WB) detection of these tagged proteins (Table 1). As shown in Figure 1B–E, we observed all expected proteins by immunoblot.

The F-box and ubiquitylation target protein (substrate) were cloned into the *pACYCDuet-1* vector for expression. The *His* and *S* tags in this vector were replaced with Myc and MBP-HA, respectively (Figure 1F), to ensure that the F-box and substrate were tagged differently from other components in the cascade. This modified plasmid was named *pACYCDuet-Myc-MBP-HA*. As shown in Figure 1G, MBP-HA can be detected using WB. The coding sequence of Myc was too short, so we did not detect it before verifying the validity of this system. However, after performing the ubiquitination cascade, the F-box fused with Myc was detected.

### 2.2. Plasmid Construction

To construct the *pCDFDuet-AtUBA1-S* plasmid, the CDS of *AtUBA1* was amplified by primers UBA1 F/R (Appendix A), and then inserted between the *Eco*RV and *Kpn*I sites of *pCDFDuet-1*.

To construct the *pCDFDuet-AtUBC8-S* or *pCDFDuet-AtUBC8-S-AtUBC1-S* plasmids, the sequence of *AtUBC8-S* was amplified by primers UBC8-S F/R (Appendix A), and then inserted into the *Eco*RI site of *pCDFDuet-1* and *pCDFDuet-AtUBA1-S*, respectively.

To construct the *pRSFDuet-His-Flag-AtUBQ11* plasmid, the sequence of HIS-*Flag-UBQ11(1-228)* was amplified by primers HIS-FLAG-UBQ F/R (Appendix A), and then inserted between the *Nco*I and *Kpn*I sites of *pRSFDuet-1*.

The construction of the *pETDuet-AtRBX1-T7-AtASK1-T7-AtCUL1-S* plasmid was performed in two steps. First, the sequence of *AtRBX1-T7-RBS-AtASK1-T7* was synthesized by Genewiz (Suzhou, China) and inserted between the *Nco*I and *Sal*I sites of *pETDuet-1*. In the second step, the CDS of *AtCUL1* was amplified by primers CUL1 F/R (Appendix A) and then inserted into the *Kpn*I site of the plasmid obtained in the first step.

To construct the *pACYCDuet-Myc-MBP-HA* plasmid, the sequence of *MCS-Myc-T7 promoter-lac operator-RBS-MBP-MCS-HA* was synthesized by Genewiz and then inserted between the *Nco*I and *Xma*JI sites of *pACYCDuet-1*.

To construct the *pACYCDuet-AtTIR1-Myc-MBP-HA* or *pACYCDuet-AtTIR1^P10A^-Myc-MBP-HA* plasmid, the CDSs of *AtTIR1* or *AtTIR1^P10A^* [20] were amplified by primers TIR1 F/R (Appendix A) and then inserted into the *Bam*HI site of *pACYCDuet-Myc-MBP-HA*.

To construct the *pACYCDuet-AtTIR1-Myc-MBP-AtIAA6-HA* and *pACYCDuet-Myc-MBP-AtIAA6-HA* plasmids, the CDS of *AtIAA6* was amplified by primers IAA6 F/R (Appendix A) and then inserted into the *Eco*147I site of *pACYCDuet-AtTIR1-Myc-MBP-HA* and *pACYCDuet-Myc-MBP-HA*, respectively.

To construct the *pETDuet-AtRBX1-T7-HsSKP1-T7-AtCUL1-S* plasmid, the sequence of *T7-RBS-HsSKP1* was synthesized by Genewiz and then inserted between the *Bam*HI and *Sgs*I sites of *pETDuet-AtRBX1-T7-AtASK1-T7-AtCUL1-S*.

To construct the *pACYCDuet-HsFBXO28-Myc-MBP-HA* or *pACYCDuet-ΔHsFBXO28-Myc-MBP-HA* plasmid, the CDS of *HsFBXO28* or *ΔHsFBXO28* [21] was synthesized by Genewiz and then inserted between the *Bam*HI and *Eco*RV sites of *pACYCDuet-Myc-MBP-HA*.

All DNA fragments were amplified using *ApexHF* HS DNA Polymerase (Cat. NO. AG12207) or *TransStart*^®^ *FastPfu* Fly DNA Polymerase (Cat. NO. AP231), which were purchased from Accurate Biotechnology (Changsha, China) and TransGen Biotech (Beijing, China), respectively. The DNA fragments obtained by PCR were ligated into the vector backbone using the *pEASY*^®^-Basic Seamless Cloning and Assembly Kit (Cat. NO. CU201), which was purchased from TransGen Biotech (Beijing, China). All gene sequences corresponding to prokaryotic proteins were sequenced by AuGCT DNA-SYN Biotechnology (Yangling, China).

### 2.3. Co-Expression of Recombinant Proteins in E. coli

The different expression vectors constructed above were individually transformed or transformed in different combinations into *Escherichia coli* BL21 (DE3) (TransGen Biotech, China, Cat. NO. CD601) and cultured in 2× Yeast extract and Tryptone (2 × YT) liquid medium containing the corresponding antibiotic at 37 °C. When the absorbance at 600 nm reached 0.5–0.6, 0.5 mM of isopropyl β-D-1-thiogalactopyranoside (IPTG) was added to induce the expression of the target protein(s), generally at 25 °C for 8-10 h with shaking at 200 rpm. IPTG (Cat. NO. I1020) was purchased from Solarbio (Beijing, China). The *E. coli* cells were collected and lysed by Scientz-IID Ultrasonic Homogenizer of Ningbo Scientz Biotechnology (Ningbo, China). Crude proteins were used to detect recombinant proteins or for ubiquitination assays.

### 2.4. SDS-PAGE and Immunoblot Analysis for Detecting the Ubiquitination Cascade

For immunoblot analysis, the crude proteins were separated on 12% SDS-polyacrylamide gels (PAGEs). The electrophoresed proteins were transferred to a 0.45 μm nitrocellulose membrane using a wet transfer system and incubated with different antibodies. Anti-S (Cat. NO. K200013M) and anti-T7 (Cat. NO. K200008M) were purchased from Solarbio (Beijing, China), anti-HA (Cat. NO. M20021) and anti-Myc (Cat. NO. M20019) were purchased from Abmart (Shanghai, China), and anti-ubiquitin (Cat. NO. AF1705) was purchased from Beyotime (Shanghai, China). The secondary antibody Goat Anti-Rabbit Mouse IgG-HRP (Cat. NO. M21003) was purchased from Abmart (Shanghai, China) and used as per the manufacturer’s instructions. Horseradish peroxidase was detected using the Super ECL Western Blotting Substrate (Cat. NO. SL1350), which was purchased from Coolaber (Beijing, China).

### 2.5. Accession Numbers

The gene sequence data from this article can be found at “https://www.ncbi.nlm.nih.gov/ (assessed on 25 September 2024)” data libraries under the following accession numbers: *AtUBA1* (NCBI Gene ID: 817562, TAIR:AT2G30110), *AtUBC8* (NCBI Gene ID: 834173, TAIR:AT5G41700), *AtUBQ11* (NCBI Gene ID: 825847, TAIR:AT4G05050), *AtRBX1* (NCBI Gene ID: 832179, TAIR:AT5G20570), *AtCUL1* (NCBI Gene ID: 825648, TAIR:AT4G02570), *AtASK1* (NCBI Gene ID: 843928, TAIR:AT1G75950), *AtTIR1* (NCBI Gene ID: 825473, TAIR:AT3G62980), *AtIAA6* (NCBI Gene ID: 841717, TAIR:AT1G52830), *HsSKP1* (NCBI Gene ID: 6500, HGNC:HGNC:10899), and *HsFBXO28* (NCBI Gene ID: 23219, HGNC:HGNC:29046).

## 3. Results

### 3.1. Auto-Ubiquitination Activity of AtSCF^TIR1^ in Recombinant E. coli

To verify that the SCF-type E3 ubiquitin ligase-mediated ubiquitination cascade can be reconstructed in bacteria, we chose the well-studied plant F-box protein AtTIR1 (Transport Inhibitor Response 1) to test our system. As receptors for the plant hormone auxin, AtTIR1 and its homologs, the AtAFBs (Auxin Signaling F-box proteins), play key roles in the auxin signaling pathway. Auxin regulates the transcription of auxin response genes through the action of TIR1/AFBs, Aux/IAA transcriptional suppressors, and ARFs (Auxin Response Factors). In general, Aux/IAAs work by binding directly to ARFs and recruiting the corepressor TPL (TOPLESS) to chromatin [22,23]. In plants, Aux/IAAs can be sensed by TIR1/AFBs, which interact with ASK1, RBX1, and CUL1 to form SCF^TIR1/AFBs^ ubiquitin ligase complexes. In the presence of auxin, the SCF^TIR1/AFBs^ ligases interact with Aux/IAAs to mediate their degradation.

Previous studies have shown that most E3 ubiquitin ligases can also undergo auto-ubiquitination when their specific substrate protein is missing [24]. Although F-box proteins are only one subunit of SCF-type E3 ubiquitin ligases, the auto-ubiquitination of F-box proteins has been demonstrated in yeast and human systems [25,26]. For the auxin receptor AtTIR1, there are no reports that it can undergo auto-ubiquitination, but Yu’s study in 2015 has shown that AtTIR1 is subject to autocatalytic degradation when assembled into an SCF complex [27]. Now, the degradation of poly-ubiquitin-tagged proteins is well known. For an SCF-type E3 ubiquitin ligase, its catalytic domain Ring-Finger is located in the RBX1 subunit. This means that the F-box protein in monomeric form is devoid of E3 ubiquitin ligase activity. If AtTIR1 fails to mediate its own degradation without assembling into the AtSCF^TIR1^ complex, it is possible that AtTIR1 undergoes auto-ubiquitination before autocatalytic degradation.

Based on the above hypothesis, we wanted to test the auto-ubiquitination activity of the AtSCF^TIR1^ E3 ubiquitin ligase using this bacterial system. As shown in Figure 2A, the vector *pACYCDuet-AtTIR1-Myc-MBP-HA* is constructed to express AtTIR1-Myc (73.3 kD). The *E. coli* BL21(DE3) strain containing multiple vectors, *pCDFDuet-AtUBC8-S-AtUBA1-S*, *pRSFDuet-His-Flag-AtUBQ11, pETDuet-AtRBX1-T7-AtASK1-T7-AtCUL1-S* and *pACYCDuet-AtTIR1-Myc-MBP-HA*, was cultured, and gene expression was induced by 0.5 mM of IPTG. As negative controls lacking E1 or E2 of the ubiquitination cascade, we also constructed the vectors *pCDFDuet-AtUBA1-S* and *pCDFDuet-AtUBC8-S* for co-transforming BL21(DE3) with the other plasmids (Figure 2B).

The total crude protein was separated by SDS-PAGE and subjected to Western Blot. The anti-Myc antibody detected AtTIR1-Myc in a ladder pattern, which implies that AtTIR1-Myc undergoes poly-ubiquitination modification (Figure 2C). The laddering pattern was also observed with the anti-ubiquitin antibody, but was not observed when any component of the ubiquitination cascade was absent (Figure 2C). These results indicate that AtSCF^TIR1-Myc^ has auto-ubiquitination activity, whose auto-ubiquitination can only be observed in the presence of other three subunits of the SCF complex (Figure 2C), which is consistent with previous studies showing that only AtTIR1 itself does not mediate its own degradation [27]. In addition, previous research has shown a specific mutation of the F-box domain of TIR1 (AtTIR1^P10A^) prevents its interaction with AtASK1 [20]. In our experiment, no auto-ubiquitination of AtTIR1^P10A^-Myc was detected when AtTIR1^P10A^-Myc replaced AtTIR1-Myc (Figure 2C), further indicating that AtTIR1-Myc cannot undergo auto-ubiquitination when it fails to assemble into the AtSCF^TIR1-Myc^ complex.

### 3.2. AtSCF^TIR1^ Can Catalyze AtIAA6 Ubiquitination in the Bacterial System

A previous study has shown that AtSCF^TIR1^ mediates AtIAA6 (a member of the Aux/IAA family) ubiquitination both in vitro and in vivo [28]. Therefore, we used the AtIAA6 as a substrate for AtSCF^TIR1^ to verify whether the bacterial system can analyze substrate ubiquitination. In order to co-express AtIAA6 with other components in the ubiquitination cascade, the vectors *pACYCDuet-Myc-MBP-AtIAA6-HA* (for the negative control) and *pACYCDuet-AtTIR1-Myc-MBP-AtIAA6-HA* were constructed (Figure 3A). The *E. coli* strain harboring AtUBA1-S, AtUBC8-S, His-FLAG-AtUBQ11, AtCUL1-S, AtRBX1-T7, AtASK1-T7, AtTIR1-Myc, and MBP-AtIAA6-HA (67.2 kD) was cultured. Since AtIAA6 interacted in an auxin-dependent manner with AtTIR1 [28], we added the naturally occurring auxin IAA (indole-3-acetic acid) when inducing gene expression with IPTG.

Then, the crude protein extracts were obtained by crushing the bacteria, separated by SDS-PAGE, and analyzed by WB with the corresponding antibodies. A laddering pattern was observed in the presence of all components when using substrate specific anti-HA, but was not observed when replacing MBP-AtIAA6-HA with MBP-HA (Figure 3B). When compared side by side, the immunoreactivity of HA is significantly stronger in the presence of IAA than in the absence of IAA (Figure 3B), coinciding with the idea that IAA can act as a molecular glue to enhance the interaction between TIR1 and AUX/IAA proteins [28,29]. It is noteworthy that, when using anti-ubiquitin for immunoblotting, protein ladders are also observed without MBP-AtIAA6-HA, because AtTIR1-Myc can undergo auto-ubiquitination, which can also be observed in the Myc-blot (Figure 3B). As expected, AtUBA1, AtUBC8, Ub, and AtSCF^TIR1^ are clearly required for AtIAA6 ubiquitination (Figure 3B). This is consistent with Winkler’s results [28], and also shows that this recombinant system can be used to analyze substrate ubiquitination, further confirming the feasibility of this recombinant system.

### 3.3. SCF Complex Composed of Heterologous Subunits Showed Ubiquitin Ligase Activity

For SCF-type multi-subunit E3 ubiquitin ligases, the Cullin subunit serves as a scaffold, while the RBX1 subunit endows the SCF complex with the ability to transfer ubiquitin from the E2 to either F-box proteins or specific substrates [10]. Cul and RBX1 are conserved in eukaryotes, and mammalian CUL1 and RBX1 can assemble with Arabidopsis ASK1 and the F-box protein to form active SCF complexes [28]. To verify that our system can also be used to study the ubiquitination cascade of SCF-type E3 ubiquitin ligases in animals, we wanted to replicate the recently reported auto-ubiquitination of the human HsFBXO28 in bacteria [21]. Since the F-BOX protein can only be recognized by the specific SKP protein [10,14], we replaced AtASK1 in the bacterial system with HsSKP1.

As shown in Figure 4A, the vector *pETDuet-AtRBX1-T7-HsSKP1-T7-AtCUL1-S* was constructed to express AtRBX1-T7, AtCUL1-S, and HsSKP1-T7 (20.5 kD). The vector *pACYCDuet-HsFBXO28-Myc-MBP-HA* was constructed to express HsFBXO28-Myc (47.3 kD). Following the study of Cai et al. [21], the deletion of the F-box domain (ΔHsFBXO28-Myc, 41.6 kD) was used as a negative control. The *E. coli* strain harboring HsSKP1-T7, HsFBXO28-Myc, and other components of the ubiquitination cascade derived from *A. thaliana* was cultured and induced. The crude protein extracts were separated by SDS-PAGE and analyzed by WB with the corresponding antibodies. A laddering pattern was observed in the presence of all components when using anti-Myc or anti-ubiquitin (Figure 4B). There was no auto-ubiquitination of ΔHsFBXO28-Myc detected when ΔHsFBXO28-Myc replaced HsFBXO28-Myc, which is consistent with previous studies by Cai et al. [21]. This result indicate that our system can also be applied to study the ubiquitination cascade of SCF-type E3 ubiquitin ligases in organisms other than plants.

## 4. Discussion

To date, countless different E3 ubiquitin ligases have been predicted in eukaryotic genomes on the basis of structural motifs. These E3s belong to different groups, among which the SCF-type E3 ubiquitin ligase group is the largest and best characterized [30]. F-box proteins confer the SCF-type ubiquitin ligases the ability to recognize target proteins, which is dependent on their C-terminal variable recruitment module. In eukaryotes, the F-box protein family is the largest known protein superfamily, and many F-box proteins with different C-terminal motifs have been identified as SCF components. The number of F-box proteins varies widely among organisms and shows little correlation with their complexity, genome size, or life cycles [31,32,33]. For example, there are 20 F-box proteins in budding yeast (*S. cerevisiae*), 27 in fruit flies (*D. melanogaster*), 69 in humans, 520 in *C. elegans*, and a staggering 897 in the model plant *A. thaliana* [31,32,33,34]. Interestingly, there is little overlap between F-box proteins across organisms, e.g., only six F-box proteins have apparent orthologs conserved between humans and flies [31,32]. Moreover, there are significant differences in F-box gene loci among closely related species. For instance, *A. thaliana* and *A. lyrata*, which diverged only five Mya, differ by 453 F-box loci, with *A. thaliana* acquiring 109 loci and losing 468 loci [33]. Current evolutionary studies speculated that huge differences in the F-box proteins between species show the rapid evolution of this superfamily, which potentially reflects a central role for ubiquitination in driving biological adaptations [33,35].

Consistent with evolutionary studies, more than 20 years of research on *A. thaliana* F-box proteins has revealed their important functions [36]. F-box proteins are critical regulators of many biological pathways, including phytohormone signaling pathways, floral development, seed germination, leaf senescence, self-incompatibility, lateral shoot branching, control of cell cycle, regulation of circadian clock, floral meristem formation, floral organ identity determination, and photomorphogenesis [36,37,38], and very much decide the fate of a plant. Although the number of F-box proteins in mammals such as humans is much smaller than in *A. thaliana*, they participate in numerous vital physiological and pathological processes, including cell cycle regulation, cell proliferation, gene expression regulation, apoptosis, and signal transduction [39]. In particular, there are concerns about the important roles of F-box proteins in carcinogenesis and tumor progression [40].

Although there are established roles for F-box proteins in many diverse pathways, the majority of the predicted F-box proteins has no known substrates. Further research on deciphering the physiological and biochemical functions of the F-box proteins in cells is absolutely necessary. Here, we present an improved, simple, and quick bacterial system for the study of SCF-type E3 ubiquitin ligases and their substrates. In vitro ubiquitination analysis using this system avoids the purification of every component in the cascade and every subunit of the SCF complex, which reduces the difficulty and increases the efficiency of the experiments. This system can also be used to obtain ubiquitination-modified target proteins in large quantities through bacterial culture, which facilitates later biochemical, biophysical, and crystallographic analyses. Furthermore, this system is expected to be used for the high-throughput screening of E3 ubiquitin ligase targets in future studies [15,17], which may boost research on the SCF complex and may help delineating F-box-substrate pairs.

In this article, we applied this system to reproduce both AtSCF^TIR1^-mediated and a mixed-species SCF^FBXO28^-mediated ubiquitylation. Special care should be taken to select the paired SKP and F-box protein when using this system, as some F-box proteins might be recognized only by specific SKPs. Our assay could also be used with E3 and substrates from other plant species. Additionally, our assay can also be applied to single-subunit E3 ligases by replacing the SCF components with a monomeric E3 ligase, while considering the replacement of suitable E2 [41].

## 5. Conclusions

In vitro ubiquitination detection of multi-subunit E3 ubiquitin ligase SCF complex has always been a challenging experiment. Here, we present an improved, simple, and quick bacterial system for the study of SCF-type E3 ubiquitin ligases and their substrates. When CUL1, RBX1, SKP1, and F-box protein are co-expressed with other components of the ubiquitination cascade, ubiquitination can occur in *E. coli* cells. By using this system, the ubiquitination cascade of AtSCF^TIR1^ and mixed-species SCF^FBXO28^ was reproduced in bacteria, which implies that this platform is generally applicable to the study of SCF complexes in eukaryotes. Using this system avoids the purification of every component in the cascade and every subunit of the SCF complex, which reduces the difficulty and increases the efficiency of the experiments.

## Figures and Tables

**Figure 1 biomolecules-14-01209-f001:**
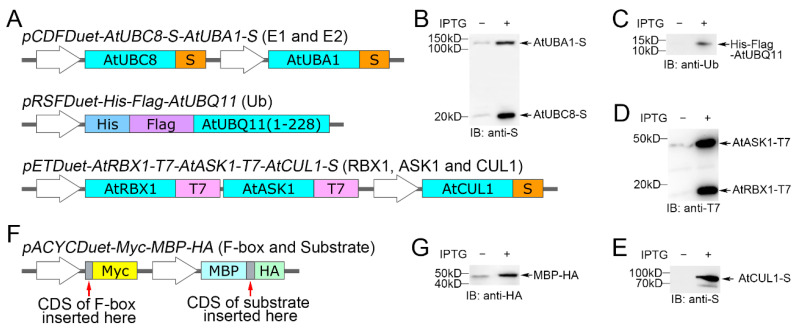
Expression of all components of the ubiquitination cascade by the *Duet* vector system. (**A**) The structures of *pCDFDuet-AtUBC8-S-AtUBA1-S*, *pRSFDuet-His-Flag-AtUBQ11*, and *pETDuet-AtRBX1-T7-AtASK1-T7-AtCUL1-S*. AtUBC8 and AtUBA1 are S-tagged at their C-termini. AtUBQ11 is His-FLAG-tagged at its N-terminus. AtRBX1 and AtASK1 are T7-tagged at their C-termini. AtCUL1 is S-tagged at its C-terminus. (**B**–**E**) The *E. coli* strains BL21(DE3) containing the individual plasmids in A were cultured separately. Crude proteins obtained by lysing these bacteria via ultrasonication were used to detect recombinant proteins using immunoblot. (**F**) The vector *pACYCDuet-Myc-MBP-HA* designed for the expression of the Myc-tagged F-box protein and MBP-HA-tagged substrate protein. (**G**) After obtaining crude proteins from the lysed bacteria containing the plasmid in F, an HA-blot is used to detect MBP-HA. In B-E and G, crude proteins from strains not induced by IPTG were used for negative controls, with some recombinant proteins showing a weak, leaky expression. (**A**,**F**) Turquoise represents CDS of different genes, orange represents S-tagged, blue represents His-tagged, purple represents Flag-tagged, pink represents T7-tagged, yellow represents Myc-tagged, cyan represents MBP-tagged, and light green represents HA-tagged. Original images can be found in Appendix A.

**Figure 2 biomolecules-14-01209-f002:**
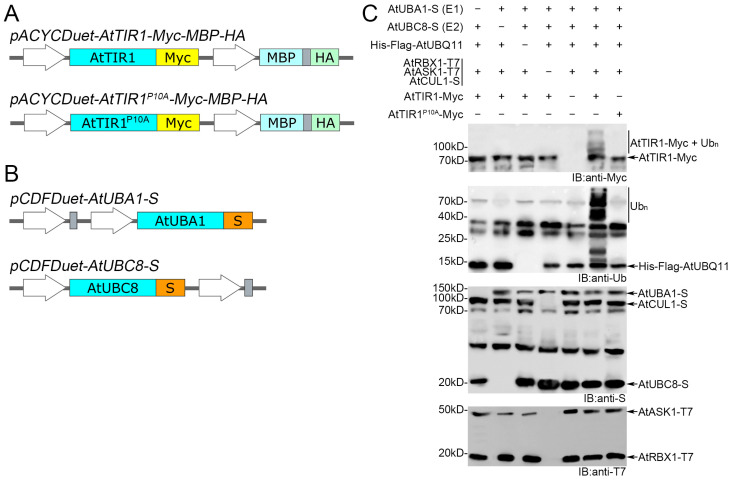
Reconstitution of AtSCF^TIR1^ auto-ubiquitination in *Escherichia coli* bacteria. (**A**) The structures of *pACYCDuet-AtTIR1-Myc-MBP-HA* and *pACYCDuet-AtTIR1^P10A^-Myc-MBP-HA*. AtTIR1 or AtTIR1^P10A^ are Myc-tagged at their C-termini. (**B**) The structures of *pCDFDuet-AtUBA1-S* and *pCDFDuet-AtUBC8-S*. AtUBA1 and AtUBC8 are S-tagged at their C-termini. (**C**) The auto-ubiquitination of AtSCF^TIR1^ is detected after the co-expression of all components of the ubiquitination cascade, except the substrate in *E coli*. The strains missing one or more of these components served as the negative controls. The mutation of AtTIR1^P10A^-Myc co-expressed with other components of the ubiquitination cascade served as another negative control. Auto-ubiquitination activities of AtSCF^TIR1^ were analyzed by Western Blot with anti-Myc or anti-ubiquitin antibodies. AtUBA1, AtUBC8, and AtCUL1 were analyzed by WB with anti-S antibodies. AtRBX1 and AtASK1 were analyzed by WB with anti-T7 antibodies. (**A**,**B**) Turquoise represents CDS of different genes, orange represents S-tagged, yellow represents Myc-tagged, cyan represents MBP-tagged, and light green represents HA-tagged. Original images can be found in Appendix A.

**Figure 3 biomolecules-14-01209-f003:**
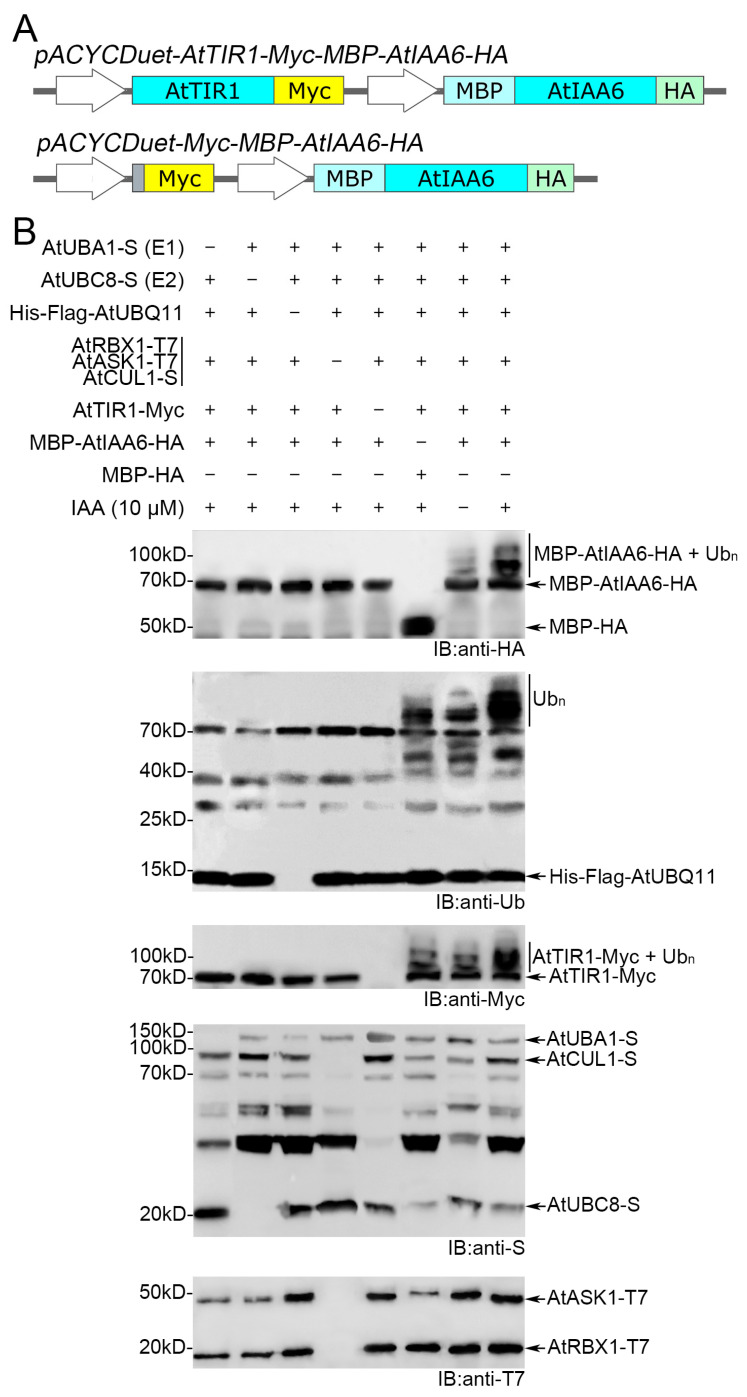
Ubiquitination of AtIAA6 by AtSCF^TIR1^ in *Escherichia coli* bacteria. (**A**) The structures of *pACYCDuet-AtTIR1-Myc-MBP-AtIAA6-HA* and *pACYCDuet-Myc-MBP-AtIAA6-HA*. AtTIR1 is Myc-tagged at its C-terminus. AtIAA6 is MBP-tagged at its N-terminus and HA-tagged at its C-terminus. Turquoise represents CDS of different genes, orange represents S-tagged, yellow represents Myc-tagged, cyan represents MBP-tagged, and light green represents HA-tagged. (**B**) The ubiquitination of AtIAA6 is detected after the co-expression of all components of the ubiquitination cascade in *E coli*. The strains missing one or more of these components served as negative controls. Auxin IAA was added to all expression systems, except to the control. The ubiquitination of AtIAA6 was analyzed by WB with anti-HA. The auto-ubiquitination of AtTIR1 was analyzed by WB with anti-Myc. The ubiquitination of AtIAA6 and AtTIR1 was detected simultaneously using anti-ubiquitin. AtUBA1, AtUBC8, and AtCUL1 were analyzed by WB with anti-S. AtRBX1 and AtASK1 were analyzed by WB with anti-T7. Original images can be found in Appendix A.

**Figure 4 biomolecules-14-01209-f004:**
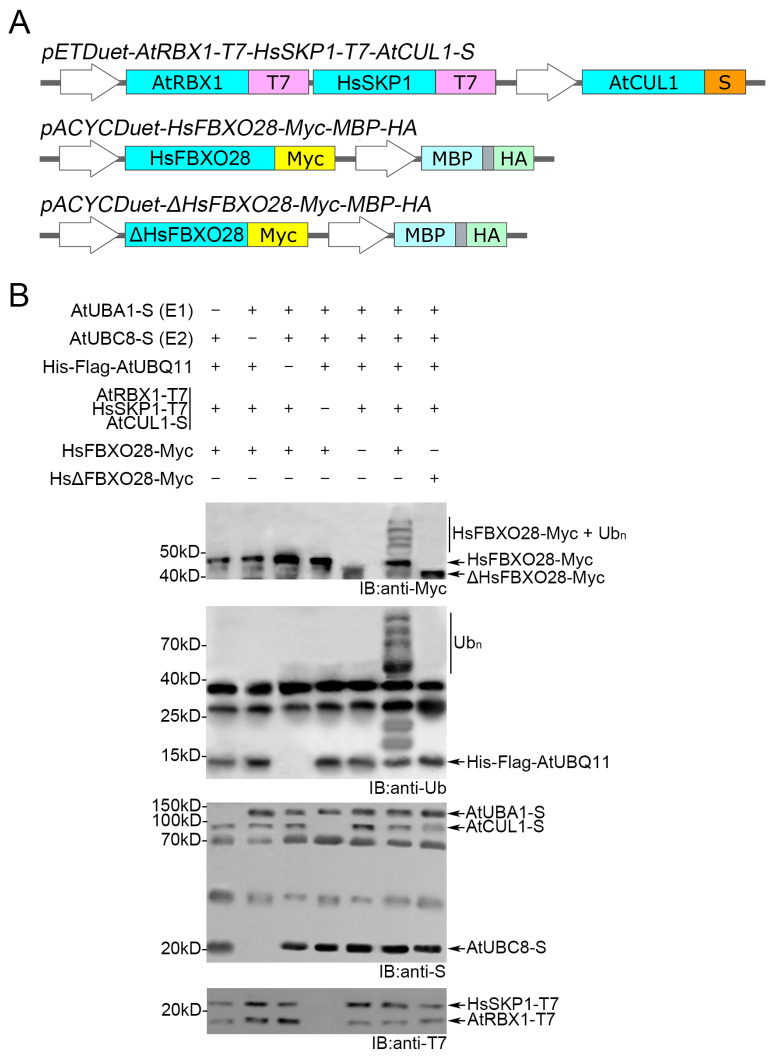
Reconstitution of SCF^FBXO28^ auto-ubiquitination in *Escherichia coli.* (**A**) The structures of *pETDuet-AtRBX1-T7-HsSKP1-T7-AtCUL1-S, pACYCDuet-HsFBXO28-Myc-MBP-HA*, and *pACYCDuet-ΔHsFBXO28-Myc-MBP-HA*. The human protein HsHFBXO28 and its inactive form ΔHsFBXO28 are Myc-tagged at their C-termini. Turquoise represents CDS of different genes, orange represents S-tagged, purple represents Flag-tagged, pink represents T7-tagged, yellow represents Myc-tagged, cyan represents MBP-tagged, and light green represents HA-tagged. (**B**) The auto-ubiquitination of SCF^FBXO28^ is detected after the co-expression of all components of the ubiquitination cascade, except substrate in *E coli*. SCF^FBXO28^ was composed of heterologous subunits: AtCUL1, AtRBX1, HsSKP1, and HsFBXO28. The strains missing one or more of these components served as negative controls. Mutated ΔHsFBXO28-Myc served as another negative control. Auto-ubiquitination activities of SCF^FBXO28^ were analyzed by WB with anti-Myc or anti-ubiquitin. AtUBA1, AtUBC8, and AtCUL1 were analyzed by WB with anti-S. AtRBX1 and HsSKP1 were analyzed by WB with anti-T7. Original images can be found in Appendix A.

**Table 1 biomolecules-14-01209-t001:** Molecular weight (MW) of proteins involved in ubiquitination cascade.

**Protein**	AtUBC8-S	AtUBA1-S	His-Flag-Ub	AtRBX1-T7	AtASK1-T7	AtCUL1-S	MBP-HA
**MW**	20.9 kD	123.9 kD	13.6 kD	17.9 kD	44.5 kD	90.8 kD	46.0 kD

## Data Availability

All data generated or analyzed during this study are available in this published article and its Appendix A. The plasmids will be available at Addgene “https://www.addgene.org/ (assessed on 25 September 2024)” or MiaoLingBio “http://www.miaolingbio.com/ (assessed on 25 September 2024)”.

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
