# Peer review of "A Bacterial Platform for Studying Ubiquitination Cascades Anchored by SCF-Type E3 Ubiquitin Ligases"

_biomolecules, 2024, doi:10.3390/biom14101209_

Round 1

Reviewer 1 Report

Comments and Suggestions for Authors

The manuscript “Bacteria Platforms for studying ubiquitination cascade of SCF-type E3 ubiquitin ligases” was submitted for review.

Ubiquitination is a fundamental principle of protein degradation, DNA repair, transcription, cell differentiation, cell cycle regulation, stress response, etc. Covalent attachment of ubiquitin to the target protein requires an ATP-dependent multi-step process controlled by ubiquitin-activating enzymes (E1), ubiquitin-binding enzymes (E2), and ubiquitin ligases (E3s), which is also known as the ubiquitination cascade. Among the four E3 groups, RING E3 is the most abundant type of ubiquitin ligase. in eukaryotes. Research to date has shown that all eukaryotes contain the Cullin family of proteins and that most Cullin protein homologs can be identified across kingdoms. When studying E3 ligases, it is necessary to detect E3 ligase autoubiquitination and substrate ubiquitination. The traditional method is to purify all components involved in the ubiquitination cascade, including E1, E2, ubiquitin, E3. The authors expressed all components of the SCF-type lig-E3 ubiquitination cascade.

This system is highly efficient and easy to operate. Here, the authors chose AtSCFTIR1 as a representative SCF-type E3 ligase and AtIAA6 as a substrate protein to evaluate this system.

The authors used modern methods for assessing indicators, including Plasmid Construction, Co-Expression of Recombinant Proteins in E. Coli, SDS-PAGE and Immunoblot Analysis for Detecting Ubiquitination Cascade. The results of the study are well presented, depicted in 3 figures and 1 table. This article presents a bacterial system for detecting the E3 ubiquitination cascade of SCF-type ubiquitin ligases, which is used in a system to reproduce AtSCFTIR1-mediated ubiquitylation of AtIAA6. The value of this method lies in its convenience, which can be used easily and quickly to detect autoubiquitination and substrate ubiquitination. This bacterial system may provide a basic platform for studying SCF-type E3 ubiquitin ligases. The abstract is well written and fully corresponds to the content of the article. The bibliography consists of 34 sources, including the latest years of publication. The article may be accepted unchanged.

Author Response

Thank you for your positive comments on our manuscript, and we have revised a few places where descriptions were not quite right in this round of revisions, as detailed in the markup in the manuscript.

Reviewer 2 Report

Comments and Suggestions for Authors

Comments on the Quality of English Language

Reviewer 3 Report

Comments and Suggestions for Authors

The manuscript entitled "Bacteria Platforms for Studying Ubiquitination Cascade of 2 SCF-Type E3 Ubiquitin Ligases" describes a method for studies of SCF-type E3 ligases in bacteria that do not contain components of the ubiquitin-proteasome system. Therefore, using the bacterial system can be considered an in-vitro experiment. Here the authors co-expressed the different components of the SCF E3 ligase as well as E1, E2, and ubiquitin and followed the ubiquitination of a model substrate.

This is an elegant method that can be useful for multiple in vitro studies of SCF proteins. However, several issues need to be clarified before concidering the manuscript for publication:

 ·      The paper doesn’t seem to be a stand-alone article but rather a technical note or Method article and therefore should be published in the correct context.

·      The introduction and discussion parts do not clarify what was already done by others to establish bacterial systems for studying the ubiquitination cascade. This includes both SCF as well as other E3 ligases.

·      The text of the abstract introduction must be improved as it contains several ambiguities. For example, the authors stated, " the detection ubiquitination cascade of that in vitro will be more complicated". This sentence is unclear. The authors go through the entire text and clarify it.

·      The first part of the results deals with vector construction. This should be moved to the "methods" section.

·      The discussion section does not place the results of the current manuscript in the context of the field. There is no discussion of what has been done thus far and how the new method can be utilized to improve our understanding of the ubiquitination cascade, mediated by SCF E3 ligases.

·      It is important that the authors examine other F-box proteins to determine the modularity of their system. 

Comments on the Quality of English Language

There are several ambiguous sentences and typos. All must be corrected

Author Response

Please see PDF files.

Round 2

Reviewer 3 Report

Comments and Suggestions for Authors

In their revised version, the authors changed some text and added substrate to test their system. The new results clearly show the universality of the bacterial ubiquitination system.  Indeed the manuscript was improved. I still have several comments:

- Bacterial ubiquitination systems had been established previously. For example see Levin-Kravets et al., 2018,  Rosenbaum J et al., 2011. It is important to present previous developments in the field in order to emphasize the new methodology for studying the SCF complex in e. coli.

- The last part of the abstract dealing with FBXO28 should be shortened to provide the general idea rather the full description of the experiment.

- The authors wrote in the rebuttal:  The last part of the abstract dealing with FBXO28 should be shortened to provide the general idea rather the full description of the experiment. I still disagree. As a technical note, in order for other researchers to be able to repeat the experiments, plasmid construction and data must be clearly separated.

- I still think that the discussion should place the assay the authors developed in the context of the state-of-the-art in the field. What is different from other methods: what are the advantages and disadvantages of the method? What other uses could be for the method? etc.

Comments on the Quality of English Language

The authors go through the entire manuscript and improve English correctness.
